# Identifying Probable Dementia in Undiagnosed Black and White Americans Using Machine Learning in Veterans Health Administration Electronic Health Records

Yijun Shao [1,2], Kaitlin Todd [3,4], Andrew Shutes-David [3,5], Steven P. Millard [3], Karl Brown [3], Amy Thomas [3,6], Kathryn Chen [7,8,9], Katherine Wilson [3,10], Qing T. Zeng [1,2] and Debby W. Tsuang [3,7,*]

1   Washington DC VA Medical Center, Washington, DC 20422, USA
2   Department of Clinical Research and Leadership, George Washington University, Washington, DC 20037, USA
3   Geriatric Research, Education, and Clinical Center, S182 GRECC, VA Puget Sound Health Care System, 1660 S. Columbian Way, Seattle, WA 98108, USA
4   Fred Hutchinson Cancer Center, Seattle, WA 98109, USA
5   Mental Illness Research, Education, and Clinical Center, VA Puget Sound Health Care System, Seattle, WA 98108, USA
6   Department of Medicine, University of Washington, Seattle, WA 98195, USA
7   Department of Psychiatry and Behavioral Sciences, University of Washington, Seattle, WA 98195, USA
8   William S. Middleton Memorial Veterans Hospital, Madison, WI 53705, USA
9   Department of Psychiatry, University of Wisconsin, Madison WI 53705, USA
10  Department of Biostatistics, University of Washington, Seattle, WA 98195, USA
*   Correspondence: dwt1@uw.edu; Tel.: +1-(206)-277-1333

**Abstract:** The application of natural language processing and machine learning (ML) in electronic health records (EHRs) may help reduce dementia underdiagnosis, but models that are not designed to reflect minority populations may instead perpetuate underdiagnosis. To improve the identification of undiagnosed dementia, particularly in Black Americans (BAs), we developed support vector machine (SVM) ML models to assign dementia risk scores based on features identified in unstructured EHR data (via latent Dirichlet allocation and stable topic extraction in n = 1 M notes) and structured EHR data. We hypothesized that separate models would show differentiation between racial groups, so the models were fit separately for BAs (n = 5 K with dementia ICD codes, n = 5 K without) and White Americans (WAs; n = 5 K with codes, n = 5 K without). To validate our method, scores were generated for separate samples of BAs (n = 10 K) and WAs (n = 10 K) without dementia codes, and the EHRs of 1.2 K of these patients were reviewed by dementia experts. All subjects were age 65+ and drawn from the VA, which meant that the samples were disproportionately male. A strong positive relationship was observed between SVM-generated risk scores and undiagnosed dementia. BAs were more likely than WAs to have undiagnosed dementia per chart review, both overall (15.3% vs. 9.5%) and among Veterans with >90th percentile cutoff scores (25.6% vs. 15.3%). With chart reviews as the reference standard and varied cutoff scores, the BA model performed slightly better than the WA model (AUC = 0.86 with negative predictive value [NPV] = 0.98, positive predictive value [PPV] = 0.26, sensitivity = 0.61, specificity = 0.92 and accuracy = 0.91 at >90th percentile cutoff vs. AUC = 0.77 with NPV = 0.98, PPV = 0.15, sensitivity = 0.43, specificity = 0.91 and accuracy = 0.89 at >90th). Our findings suggest that race-specific ML models can help identify BAs who may have undiagnosed dementia. Future studies should examine model generalizability in settings with more females and test whether incorporating these models into clinical settings increases the referral of undiagnosed BAs to specialists.

**Keywords:** electronic health record; dementia; machine learning; underdiagnosis; Veterans Health Administration

## 1. Introduction

Alzheimer's disease (AD) and related dementias (ADRD) are fatal neurodegenerative disorders, yet nearly half of those affected by ADRD have not been formally diagnosed [1,2]. This crisis of underdiagnosis exacerbates existing disparities in health care, as dementia underdiagnosis may disproportionately affect Black Americans (BAs) [3]. In a large 2019 study of Medicare claims, older BAs with dementia were about two times less likely to be correctly diagnosed with dementia than older White Americans (WAs) with dementia [3]. And, in one of the small handful of studies that examine racial disparity in dementia care within the Veterans Health Administration (VHA) [4,5], significantly fewer BA Veterans with suspected dementia underwent neuropsychological testing for the diagnosis of dementia than WA Veterans with suspected dementia [5]. The underdiagnosis of dementia translates into missed opportunities to treat patients [6], improve quality of life (e.g., through medication management and referrals) [7,8], reduce patient and family burden [9,10] and reduce hospitalization, institutionalization and health care costs [11,12].

We seek to use natural language processing (NLP) and machine learning (ML) tools to address the magnitude of dementia diagnostic disparity in the VHA Corporate Data Warehouse (CDW), which is an ideal setting for this work, as it contains comprehensive structured and unstructured data on ~0.4 million BA Veterans who are age 65+ and receive care as part of the largest integrated health care system in the nation. ML methods have previously been applied to electronic health records (EHRs) [13,14], but we have developed one of the first ML models to increase the sensitivity of potential dementia identification by using both structured EHR data (e.g., demographics, diagnoses [ICD codes], procedures [CPT codes], medications and clinical note types) *and* unstructured EHR data (e.g., words in clinical notes) [15]. In our previous work, we applied topic modeling and logistic regression to develop risk scores for dementia based on the EHRs of older Veterans with (n = 1861, mean age 79.8) and without (n = 9305, mean age 79.5) ICD-9 dementia codes who were seen in specialty clinics [15]. Here, we seek to extend this work and to improve upon the potential identification of undiagnosed dementia, particularly in BA Veterans, by developing support vector machine (SVM) models.

SVM models are useful in this context because they can efficiently accommodate a large number of features from a large, nonlinear dataset. Such an approach is advantageous because it can address classification and regression problems, avoid over-fitting the models and achieve similar performance to a random forest model but with better stability [16,17]. We apply this SVM approach separately for BAs and WAs using a larger sample of VA patients who are 65+ years old with and without ICD 9/10 diagnosed dementia. We validate these models by having dementia experts who are blinded to the dementia risk scores perform chart reviews for a new set of patients who lack ICD-9/10 dementia diagnoses and who were not used to build the models; we then compare the chart review diagnoses to the model-generated risk scores.

## 2. Materials and Methods

### 2.1. Study Population

After receiving IRB approval, we created a cohort of cases (i.e., Veterans with an ICD-9/10 dementia code) and controls (Veterans without any ICD-9/10 dementia codes) from the CDW by selecting patients who turned age 65 between 1999 and 2018, who were previously evaluated at a VA clinic and who were identified as BA or WA in their EHRs (see Figure 1). The selected Veterans were followed until 12 September 2018, until diagnosis (cases) or until censoring due to an absence of records (controls).

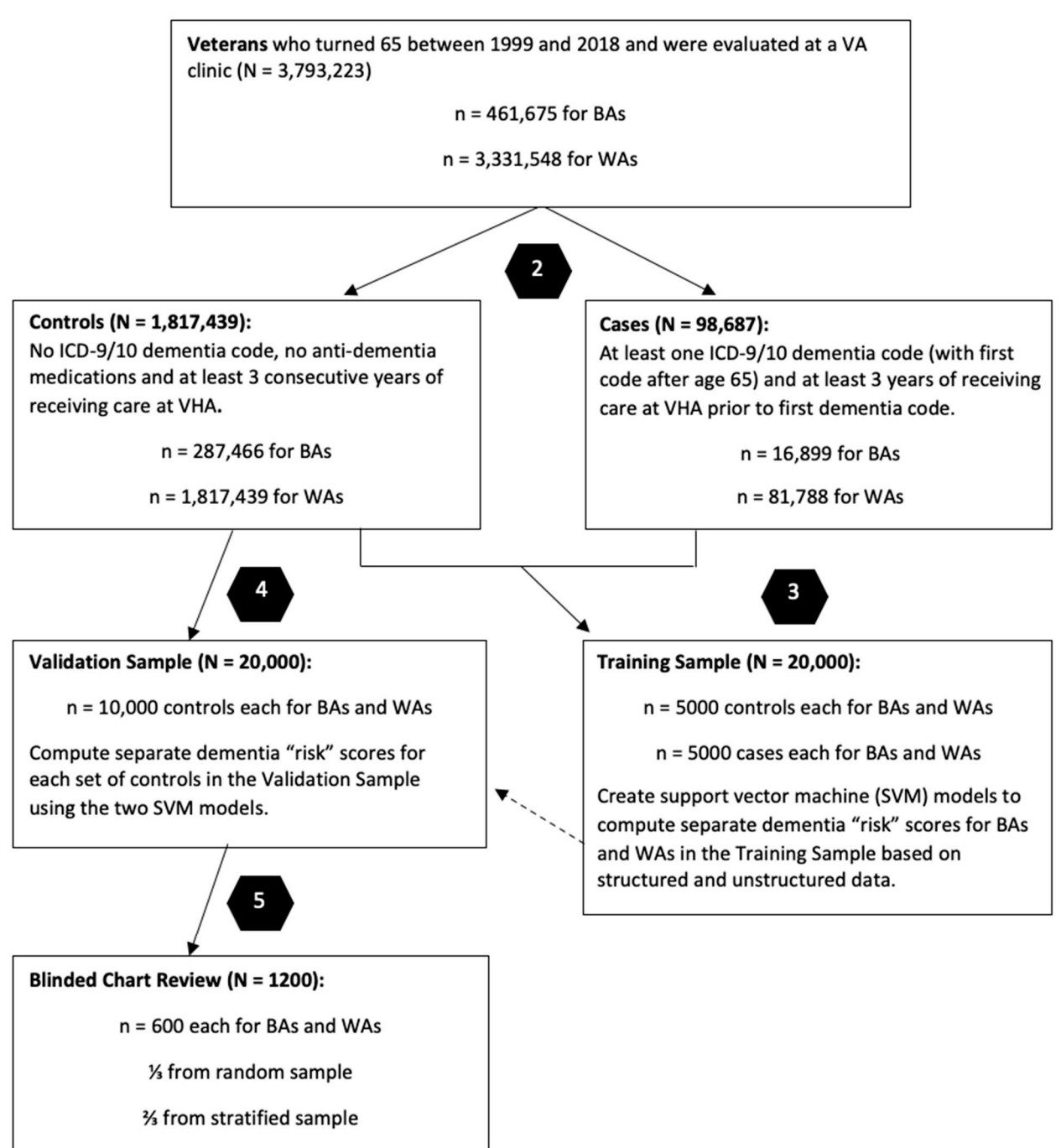

**Figure 1.** Study flow diagram for Black American (BA) and White American (WA) Veterans. The figure shows the number of Veterans available within the Veterans Health Administration (VHA) Corporate Data Warehouse (CDW) for the period under study who met inclusion/exclusion criteria (Steps 1 and 2), as well as the number of Veterans used for model building (Step 3) and validation (Step 4). Veterans in the Training Sample and Validation Sample were chosen with simple random sampling. Veterans who underwent chart review (Step 5) were chosen from the Validation Sample via simple random sampling and stratified random sampling, where the strata were based on scores. Solid lines indicate filtration/sampling; the dashed line indicates our use of the model created in Step 3 to produce "risk" scores for Veterans in the validation sample.

To accommodate our focus on late-onset dementia, cases had to have received a first diagnosis of ICD-9 or ICD-10 dementia after age 65 (with associated notes); they also had to have 2 or more other documented clinical visits (with associated notes) in each

of the 3 years prior to the first diagnosis. Conversely, controls could not have had any ICD-9/10 dementia codes; could not have filled donepezil, galantamine, rivastigmine or memantine prescriptions; and needed 3+ consecutive years with 2 or more clinical visits (with associated notes) after reaching age 62. We created separate BA and WA cohorts of cases and controls to satisfy these criteria (see Figure 1).

Clinical data were collected from EHRs for a 3-year period that either immediately preceded but did not include the first ICD-9/10 diagnosis of dementia (for cases) or a random visit date that was selected as an index date (for controls). This 3-year period was established to provide an adequate quantity of structured and unstructured data.

The sampling and modeling of the Training and Validation Samples were performed separately for BAs and WAs. We created model Training Samples by randomly sampling 5000 cases and 5000 controls for each race (total n = 20,000). For each control, we randomly chose the index visit among all visits that satisfied the 3-year lookback criterion. We used the Training Samples to build models that produced dementia risk scores. We then created model Validation Samples by randomly sampling 10,000 controls for each race who were not included as part of the Training Samples (total n = 20,000), and we used the models to generate scores for these samples. Finally, we sampled 600 Veterans from the Validation Samples for each race to undergo blinded chart reviews (total n = 1200). Veterans were selected for chart reviews via simple random sampling (n = 200) and stratified random sampling (n = 400) based on the percentiles of the full Validation Sample risk scores, such that 100 Veterans from the >75–90th percentiles were included and 30 Veterans in each of the 10 remaining upper percentile ranges (i.e., 30 each from the >90th–≤91st, >91st–≤92nd, etc.) were included.

*2.2. Variable Creation*

2.2.1. Structured Data

Structured data were treated as candidate binary variables, such that, during the 3-year period, *0* indicated an absence of the codes/medications/note type and *1* indicated their presence. To account for a transition within the VHA during the study period from the ICD-9 to ICD-10 diagnostic codes, we performed equivalence mapping, visualizing the CDC/CMS general equivalence mappings (GEM) as a large bipartite graph that consisted of two disjointed sets of vertices representing all the ICD-9 and ICD-10 codes, respectively, and a number of edges connecting ICD-9 vertices to ICD-10 vertices, representing the possible conversions from ICD-9 codes to ICD-10 codes. These mappings allowed us to decompose the GEM, viewed as a large bipartite graph, into a number of smaller disjoint bipartite subgraphs that could not be decomposed into smaller disjoint subgraphs without breaking edges. Then, for each of these minimal equivalence mappings, a new code was defined to represent the group of ICD-9 codes before the transition date and the group of ICD-10 codes after the transition. Variables corresponding to the new codes were defined in the same way as other codes (e.g., CPT codes).

Separately for BAs and WAs, we selected variables from the structured data that corresponded to the codes/medications/note types that were present in 10+ Veterans in the Training Sample. All the stable topic variables and two demographic variables (age and sex) were then used in the models.

2.2.2. Unstructured Data

Unstructured data were handled using the two-step topic modeling approach previously described by Shao et al. [15,18]. This unsupervised ML method identifies shared topics from a large text corpus. Each topic is defined as a binary variable indicating the presence or absence of that topic, and the proportion of topics within any particular document is calculated. Here, to identify dementia-related signs, we used the proportion of dementia-related topics observed in excess in cases versus controls.

More specifically, raw topics were identified in clinical notes by running a latent Dirichlet allocation (LDA) algorithm within the Machine Learning for Language Toolkit Java package [15,18], which includes topic learning and inference functions. The learning function is a time-consuming algorithm that learns the topics from a set of text documents and generates a topic model, whereas the inference function runs much faster and can apply the learned topic model to a new set of text documents and then infer the topic distributions in those documents. For our topic learning subset, we randomly sampled one note per day for each subject from the ~5 million notes collected during the 3-year study period, yielding a sample corpus of 1.8 million notes. We then randomly selected 1 million notes from this sample corpus, which allowed for a reduced running time for topic learning while ensuring that main topics were preserved. We next ran LDA topic learning 3 times on the 1 million sampled notes, setting 1000 as the total number of topics; we applied the 3 resulting models to all 5 million notes, using the topic inference function to infer the topic proportions in each note. Based on the inferred topic proportions, we calculated the number of words that were associated with each topic in each note by multiplying the topic proportion by the total number of words in the note. Because the "number of words" associated with a topic was not always a whole number, we call it the pseudo word count (PWC).

We then applied the stable topic extraction method [15,18], which yielded 852 stable topics. For each stable topic, there were 3 topics—one from each run—that were very similar to each other, and the stable topic was the "average" of the 3 similar topics. Likewise, the PWC for the stable topic in each note was defined to be the median value of the 3 PWCs that corresponded to the 3 topics (i.e., one from each run). By design, topic proportions are always positive numbers, so the PWCs are positive as well. However, because not all topics are present in every note, we set a nonzero threshold for the PWCs to indicate whether a topic was present in a note. Empirically, we set the threshold at 2.0, which roughly means that a topic is present in a note only when the PWC $\geq$ 2.0. To allow various degrees of topic presence, we defined topic presence to be a function of PWC as follows: (1) presence = 0 if PWC < 2.0, (2) presence = PWC/10.0 if $2.0 \leq$ PWC $\leq 10.0$ and (3) presence = 1.0 if PWC > 10.0. For the ML model, stable topics were used as variables/features, and the maximum presence value over all notes of each Veteran was defined as the Veteran's topic presence value.

### 2.3. Support Vector Machine (SVM) Model

Separately for BA and WA Veterans in the Training Sample, we constructed SVM models that used the selected predictor variables to generate dementia "risk" scores. To construct the SVM models, we used the linear SVM model (LinearSVC algorithm) in Python package Scikit-Learn [19]. The SVM models had only one important hyperparameter: "C", the cost parameter, which sets the trade-off between misclassification and the simplicity of the decision surface. To determine the best value for C, we performed five-fold cross-validation on the training dataset with various values for C and then selected the value corresponding to the highest predictive area under the receiver operating characteristic (ROC) curve (AUC) in the five-fold cross-validation. The selected C value was used to train the final SVM model on the entire training dataset. The linear SVM model output scores represent the distance to the separation hyperplane in the high-dimensional feature space. The scores have no theoretical limits, and higher scores indicate a higher likelihood of having dementia.

### 2.4. Validation of the SVM Model

We separately generated scores for BA and WA controls in the Validation Sample, and then, in a subset of these Veterans, we performed chart reviews in which reviewers were blinded to dementia "risk" score. Chart reviews were conducted by 3 experienced cognitive

disorder experts (2 trained in geriatric psychiatry [DWT and KC] and 1 in geriatric medicine [AT]) who achieved interrater reliability on dually reviewed charts (Cohen's Kappa value of 0.74 [se = 0.25, 95% CI = 0.25–1; $p$ = 0.0016]). The reviewers retroactively applied the DSM-V criteria for major neurocognitive disorder [20] by evaluating memory, apraxia, aphasia, agnosia, executive functioning and functional domains of ADL and iADL [21] in abstracted notes. The reviewers avoided attributing cognitive or functional deficits due to physical limitations or acute or chronic medical conditions to dementia. When reviewers were uncertain about a Veteran's dementia status, that Veteran was labeled "uncertain", and then one of the other reviewers adjudicated dementia status independently of the initial reviewer. Dementia status was coded by reviewers as "None", "Possible" or "Probable"; a probable or possible dementia code thus indicated that a Veteran likely had dementia symptoms that had not been worked up or assigned a dementia diagnosis. Using chart reviews as the reference standard, we assessed the prevalence of undiagnosed possible/probable dementia and assessed the sensitivity, specificity, positive predictive value (PPV), negative predictive value (NPV), accuracy and AUC by varying the cutoff score for determining when to declare "possible or probable undiagnosed dementia". Estimates were computed using inverse probability weighting to account for stratified sampling [22], and confidence intervals were computed using bootstrapping. Demographics, estimates and confidence intervals were computed using R [23]. We created scatter plots of dementia risks for 3 groups (probable, possible and none) as well as 2 groups (probable/possible combined and none).

## 3. Results

### 3.1. Demographics

Among the Veterans who met the inclusion/exclusion criteria (see Figure 1), the prevalence of ICD dementia was 5.5% for BAs and 4.3% for WAs. Veterans ranged in age from 65 to 84 (see demographics in Table 1). In the Training Sample, cases were older compared to controls (mean [SD] = 72.4 [4.8] vs. 69.1 [3.7]), and both cases and controls were overwhelmingly male (97.7% and 97.2%). BA Veterans were similar in mean age to WA Veterans (72.1 [4.8] vs. 72.8 [4.8] for cases; 68.6 [3.5] vs. 69.5 [3.8] for controls). The demographics for controls in the Validation and Training Sample were similar.

**Table 1.** Demographics of the Training Sample and Validation Sample by Race (BA: Black American; WA: White American).

| | Training Sample | | | | | |
| --- | --- | --- | --- | --- | --- | --- |
| | Cases (n = 10 K) | | | Controls (n = 10 K) | | |
| | BA (n = 5 K) | WA (n = 5 K) | Combined (n = 10 K) | BA (n = 5 K) | WA (n = 5 K) | Combined (n = 10 K) |
| Age, mean (SD) | 72.1 (4.8) | 72.8 (4.8) | 72.4 (4.8) | 68.6 (3.5) | 69.5 (3.8) | 69.1 (3.7) |
| Age category, (%) | | | | | | |
| 65–69 | 35.7 | 29.8 | 32.8 | 70.5 | 60.0 | 65.2 |
| 70–74 | 34.1 | 34.4 | 34.3 | 22.1 | 28.4 | 25.3 |
| 75–79 | 20.9 | 24.2 | 22.6 | 5.8 | 9.2 | 7.5 |
| 80–84 | 9.4 | 11.5 | 10.4 | 1.6 | 2.5 | 2.0 |
| Gender, % male | 97.9 | 97.5 | 97.7 | 96.8 | 97.6 | 97.2 |

**Table 1.** *Cont.*

| | Full Validation Sample (n = 20 K) | | | Chart Review Sample (n = 1200) * | | | | | |
|---|---|---|---|---|---|---|---|---|---|
| | | | | Unweighted | | | Weighted † | | |
| | BA (n = 10 K) | WA (n = 10 K) | Combined (n = 20 K) | BA (n = 600) | WA (n = 600) | Combined (n = 1200) | BA (n = 600) | WA (n = 600) | Combined (n = 1200) |
| Age, mean (SD) | 68.5 (3.4) | 69.5 (3.8) | 69.0 (3.6) | 69.3 (4.2) | 70.2 (4.5) | 69.8 (4.3) | 68.5 (3.4) | 69.3 (3.7) | 68.9 (3.6) |
| Age category, (%) | | | | | | | | | |
| 65–69 | 70.9 | 60.3 | 65.6 | 64.8 | 52.8 | 58.8 | 70.4 | 60.9 | 65.7 |
| 70–74 | 22.3 | 28.6 | 25.5 | 22.3 | 28.5 | 25.4 | 23.2 | 28.4 | 25.8 |
| 75–79 | 5.5 | 8.7 | 7.1 | 9.3 | 13.7 | 11.5 | 5.4 | 8.2 | 6.7 |
| 80–84 | 1.3 | 2.4 | 1.9 | 3.5 | 5.0 | 4.3 | 1.0 | 2.5 | 1.7 |
| Gender, % male | 96.1 | 97.6 | 96.8 | 97.2 | 97.0 | 97.1 | 96.7 | 98.9 | 97.8 |

* Patients who underwent chart review were a subset of the full Validation Sample, selected via a combination of random and stratified sampling, as described in the text. † Observations were weighted according to the inverse probability of being sampled from the full Validation Sample.

### 3.2. Variable Selection for the SVM Model

For the model trained on BA Veterans, a total of 8221 features were selected, including 2 demographics, 854 topics, 2229 nondementia ICD code groups, 2561 CPT codes, 686 medications and 1889 note types. For the model trained on WA Veterans, a total of 7716 features were selected, including 2 demographics, 854 topics, 2141 nondementia ICD code groups, 2330 CPT codes, 655 medications and 1734 note types.

The most significant topic features are shown in Supplemental Table S1. Note that the terms in a topic can occur in any order or combination, and the presence of a topic in a document does not require that all the terms in a topic be present. Topics that were observed more frequently in cases than in controls were considered dementia-related.

### 3.3. Distribution of Scores

In the Training Sample, cases had higher dementia "risk" scores than controls (mean [SD] = 0.56 [0.54] vs. −0.50 [0.36] for BAs, and 0.54 [0.55] vs. −0.47 [0.34] for WAs; Figure 2, Supplemental Figure S1). In the Validation Sample, Veterans that chart reviewers labeled as possible/probable dementia had higher scores compared to Veterans labeled as no dementia (0.45 [0.38] vs. −0.02 [0.51] for BAs, and 0.38 [0.41] vs. −0.02 [0.47] for WAs; Figure 3). For our chart review subsample of the Validation Sample, we oversampled Veterans with higher scores (i.e., Veterans with chart reviews had higher scores compared to all Validation Veterans: 0.05 [0.52] vs. −0.45 [0.41] for BA Veterans, and 0.02 [0.48] vs. −0.44 [0.38] for WA Veterans; Supplemental Figure S2), and therefore, we adjusted scores using inverse probability weighting to account for stratified sampling.

### 3.4. Prevalence of Undiagnosed Dementia and Screening Test Characteristics

Of the 1200 Veterans who underwent chart review, 15.3% (n = 92) of BAs and 9.5% (n = 57) of WAs were characterized as possible/probable dementia by the reviewers. After adjusting for stratified sampling that intentionally oversampled Veterans with higher scores, the estimated prevalence of undiagnosed possible/probable dementia in the full Validation Sample was 4.1% [3.2, 6.2] for BA Veterans and 3.6% [2.3, 6.3] for WA Veterans. There was a strong positive relationship between risk scores and the prevalence of undiagnosed possible/probable dementia (Figure 4), and as anticipated, for Veterans with scores below the 90th percentile, the percentages of undiagnosed possible/probable dementia were low: 3.9% (95% CI [2.1, 7.0]) and 2.9% (95% CI [1.3, 5.8]) for BA and WA

Veterans, respectively. Among Veterans with scores above the 90th percentile, we found that a higher percentage of BA Veterans had undiagnosed possible/probable dementia than WA Veterans: 25.6% (95% CI [20.9, 30.8]) vs. 15.3% (95% CI [11.6, 19.8]).

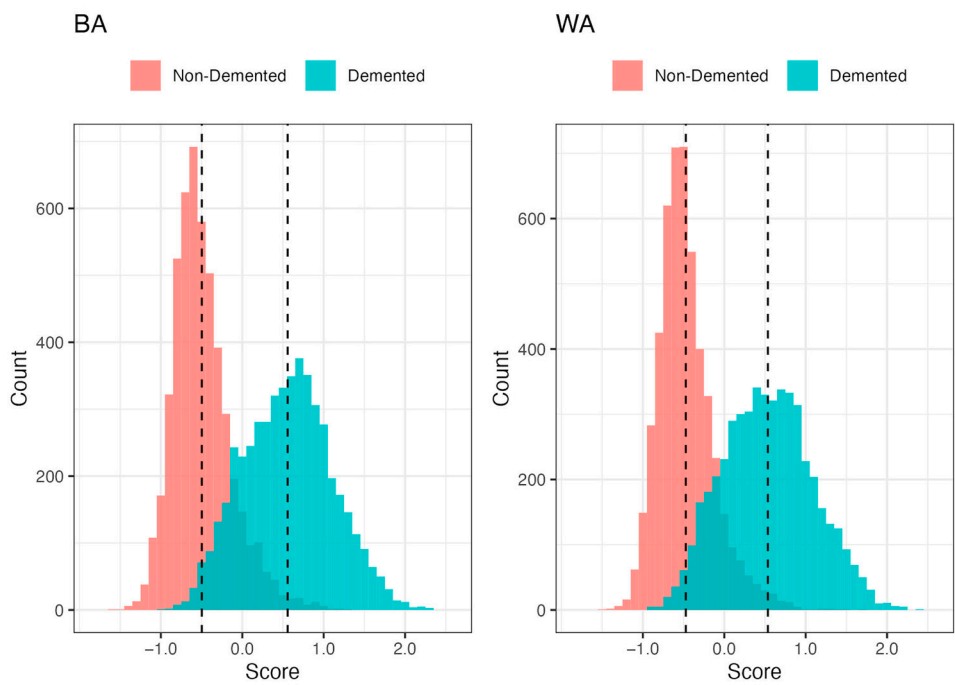

**Figure 2.** Distribution of scores by dementia status and race (BA: Black American; WA: White American) for Veterans in the Training Sample (n = 5000 in each dementia status group for each race). Dashed lines represent the means of the distribution.

Table 2 shows the sensitivity, specificity, PPV, NPV and accuracy of the screening tests for various cutoff scores. As shown in Supplemental Figure S3, the AUC was moderately high for both BA Veterans (0.86 [0.59, 0.95]) and WA Veterans (0.77 [0.59, 0.90]). For score cutoffs above the 50th percentile in the Validation Sample, sensitivity was moderate, and specificity was very high for both BA and WA Veterans (e.g., using the 90th percentile as the cutoff, sensitivity and specificity were 0.61 [0.40, 0.76] and 0.92 [0.91, 0.92], respectively, for BA Veterans, and 0.43 [0.24, 0.67] and 0.91 [0.91, 0.92], respectively, for WA Veterans). Because of the low prevalence of undiagnosed possible/probable dementia in the full Validation Samples, as well as the low sensitivity and high specificity of the screening tests, it was unsurprising that PPV was low and NPV was high [24]; using the 90th percentile as the cutoff, PPV was only 0.26 [0.21, 0.30] and 0.15 [0.12, 0.20] for BA and WA Veterans, respectively. In contrast, NPVs remained quite high regardless of the score cutoff. Accuracy improved with increasing cutoff scores; using the 90th percentile as a cutoff, accuracy was 0.91 [0.89, 0.92] and 0.89 [0.87, 0.91] for BA and WA Veterans, respectively.

**Table 2.** Values of sensitivity, specificity, positive predictive value (PPV), negative predictive value (NPV) and accuracy at various score cutoffs. Values in brackets denote 95% confidence intervals *.

| Race [†] | Cutoff Percentile | Sensitivity | Specificity | PPV | NPV | Accuracy |
|---|---|---|---|---|---|---|
| BA | 50 | 0.89 [0.53, 1] | 0.54 [0.42, 0.54] | 0.08 [0.06, 0.10] | 0.99 [0.93, 1] | 0.55 [0.49, 0.60] |
| | 75 | 0.89 [0.53, 1] | 0.79 [0.76, 0.77] | 0.15 [0.12, 0.19] | 0.99 [0.95, 1] | 0.79 [0.77, 0.80] |
| | 90 | 0.61 [0.40, 0.76] | 0.92 [0.91, 0.92] | 0.26 [0.21, 0.31] | 0.98 [0.96, 0.99] | 0.91 [0.89, 0.92] |
| | 95 | 0.37 [0.24, 0.48] | 0.97 [0.96, 0.97] | 0.31 [0.25, 0.39] | 0.97 [0.95, 0.98] | 0.94 [0.92, 0.95] |

**Table 2.** *Cont.*

| Race [†] | Cutoff Percentile | Sensitivity | Specificity | PPV | NPV | Accuracy |
|---|---|---|---|---|---|---|
| WA | 50 | 0.86 [0.47, 1] | 0.51 [0.45, 0.57] | 0.06 [0.04, 0.11] | 0.99 [0.94, 1] | 0.52 [0.46, 0.58] |
| | 75 | 0.58 [0.31, 0.85] | 0.76 [0.76, 0.77] | 0.09 [0.06, 0.12] | 0.98 [0.93, 0.99] | 0.75 [0.73, 0.77] |
| | 90 | 0.43 [0.24, 0.67] | 0.91 [0.91, 0.92] | 0.15 [0.12, 0.20] | 0.98 [0.94, 0.99] | 0.89 [0.87, 0.91] |
| | 95 | 0.30 [0.16, 0.48] | 0.96 [0.96, 0.96] | 0.22 [0.16, 0.29] | 0.97 [0.94, 0.99] | 0.94 [0.91, 0.95] |

* Estimates were computed using inverse probability weighting to account for stratified sampling, and confidence intervals were computed using bootstrapping. [†] BA: Black American; WA: White American.

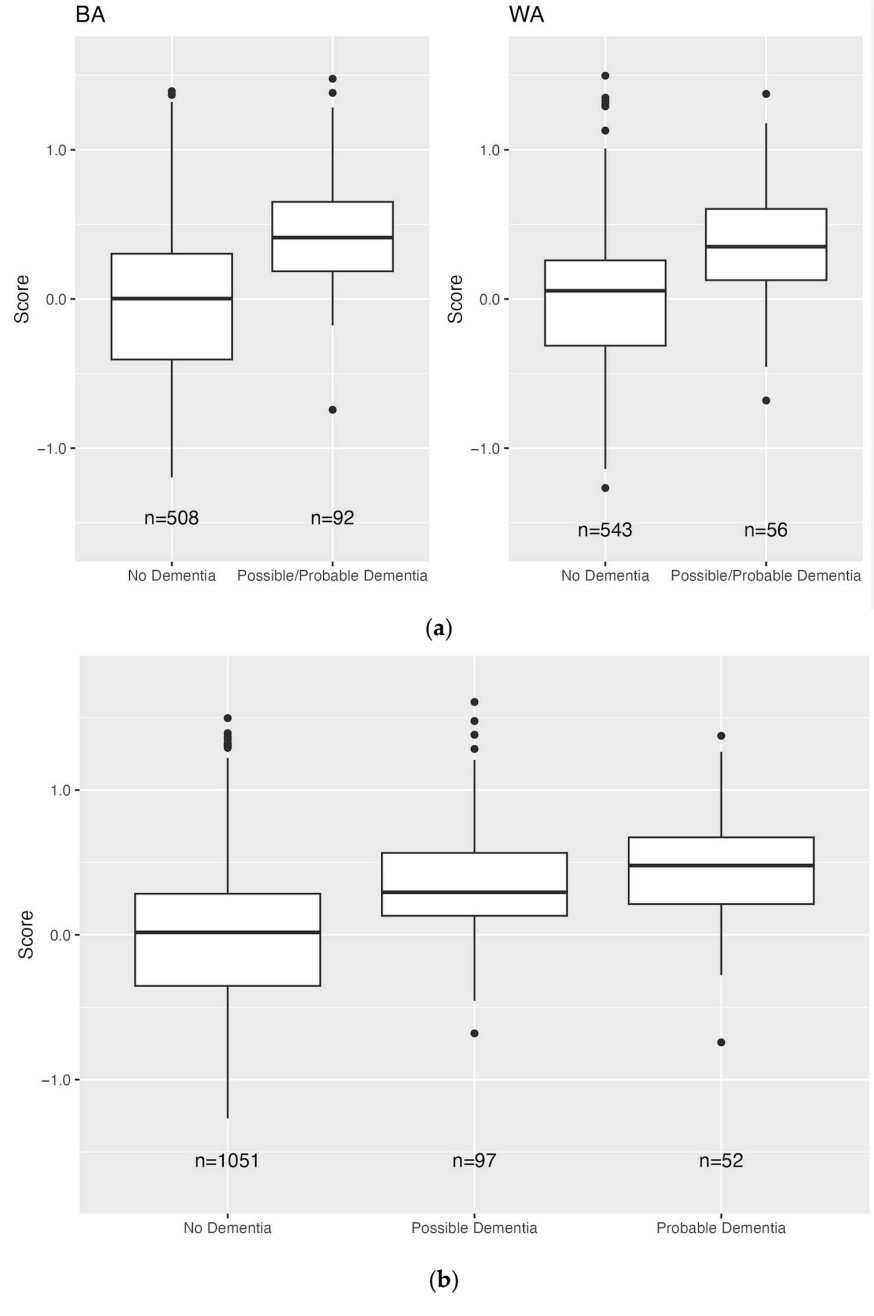

**Figure 3.** (**a**). Distribution of risk scores by dementia status and race (BA: Black American, WA: White American) for Veterans in the Validation Sample who underwent chart review (n = 600 for each race). (**b**). Distribution of risk scores by dementia status for both Black American and White American Veterans in the Validation Sample who underwent chart review (n = 600 for each race).

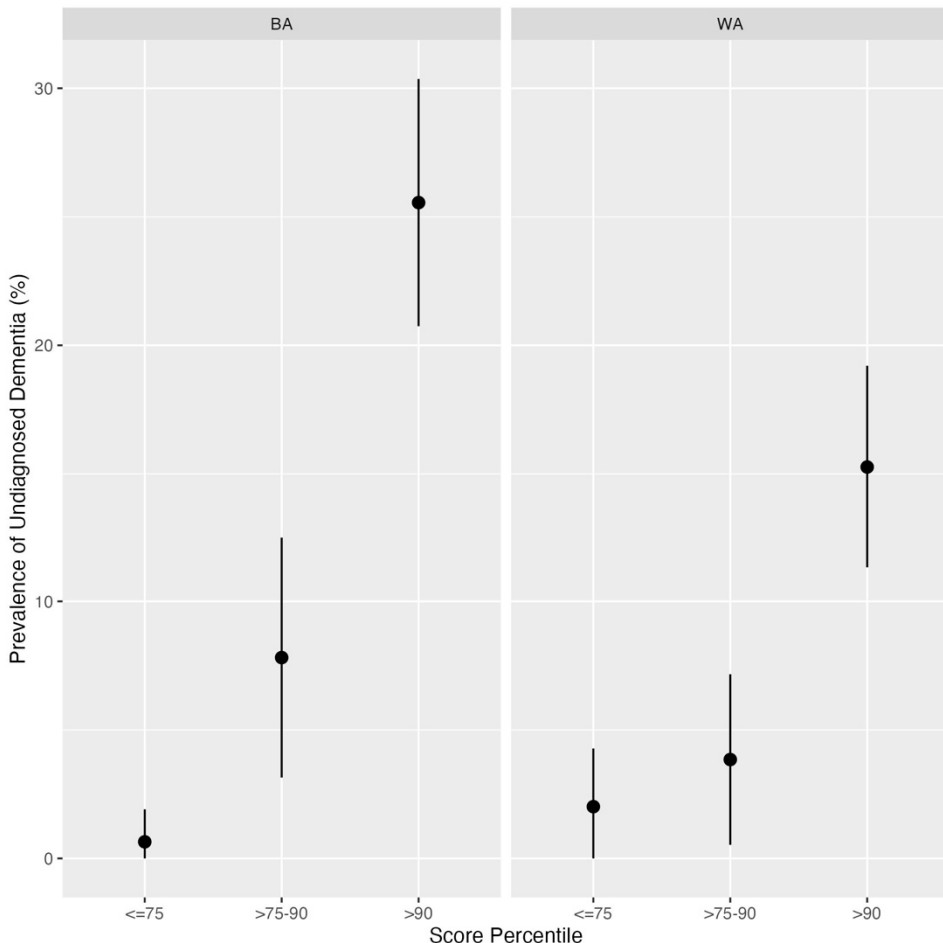

**Figure 4.** Prevalence of undiagnosed dementia by score percentile stratum and race (BA: Black American; WA: White American) for Veterans who underwent chart review (n = 600 for each race). For each race, score percentiles are based on using the scores from all 10,000 Veterans in the Validation Sample.

## 4. Discussion

### 4.1. Significance

We successfully developed and validated separate ML models to identify probable dementia cases in BA Veterans without ICD diagnoses and in WA Veterans without ICD diagnoses. The dementia risk scores generated by the SVM models were positively correlated with the diagnosis of dementia and achieved a high AUC (0.86 [0.59, 0.95]) for BA Veterans and a satisfactory AUC for WA Veterans (0.77 [0.59, 0.90]). Given that BAs are about twice as likely to develop dementia as WAs [25,26], the good performance of the SVM in this population is particularly important.

### 4.2. Context

Our preliminary data suggest that BA Veterans have different risk factors for developing dementia than WA Veterans. Using logistic regression to investigate risk factors for incident dementia in all VHA, we identified different risk factors in older BA and WA Veterans [27]. For example, among the key baseline characteristics that were significant predictors of dementia in both races, stroke was a significantly stronger predictor among BAs, and Hispanic ethnicity and depression were significantly stronger predictors among WAs ($p < 0.0001$). Those findings motivated the development of the race-specific risk models proposed in the current study, which instead focuses on prediction.

Many studies have applied NLP and ML methods in dementia [28], particularly in the context of neuroimaging [29,30] or in the use of EHRs to identify cognitive impairment or diagnosed dementia [31,32], yet few studies have sought to use EHRs as a direct phe-

notyping tool to assist in the identification of undiagnosed dementia. Researchers in the UK developed models (including SVM) to identify patients who may have dementia [33], and Kaiser Permanente/UCSF researchers developed the eRADAR tool in research participants and then validated it in two health care systems [1,34]; both studies limited their EHR interrogations to structured data and showed some success in identifying patients who may have had undiagnosed dementia. Likewise, Yadgir et al. used ML to identify structured variables associated with cognitive impairment in ER patients [35]. Conversely, Boustani et al. developed passive digital signatures that may identify ADRD by searching for predetermined variables in *both* structured and unstructured EHR data, and their work suggests that the combination can improve AUC by up to 0.11 [36]; however, like Barnes et al., Boustani et al. used curated, preselected search terms rather than leveraging the potential of NLM and ML tools to identify topic features associated with dementia.

Rather than employing a targeted-word study design like Barnes et al. or Boustani et al., we sought to improve the identification of patients who may have dementia by combining supervised ML with an improved clinical standard. More specifically, we sought to improve upon EHR ICD codes as the basis for ML by incorporating chart reviews by reviewers who were blinded to the initial ML-derived dementia likelihood scores. We previously published a ML logistic regression model that used this approach on a smaller scale, applying supervised ML to structured and unstructured data from EHRs to identify topics associated with dementia and to then identify patients who may have had undiagnosed dementia [15]. That study included blinded manual reviews for a much smaller sample (n = 140) than that of our current work, and it produced a sensitivity of 0.825 and a specificity of 0.832. It also had older Veterans (i.e., an average age of 80 vs. 71 in this study); complications with controls in the logistic regression model; and an ad hoc stratification method for computing sensitivity and specificity. Here, in the current study, our SVM models avoided these idiosyncrasies in a much larger (n = 1200) and more diverse (600 BA and 600 WA) validation effort.

EHR tools and ML models that do not specifically attempt to reflect minoritized communities are more likely to unintentionally generate cycles of exclusion and to thereby perpetuate underdiagnosis in BAs rather than addressing underdiagnosis [37]. To our knowledge, the present study is the first effort to develop and evaluate a model that specifically focuses on BAs.

*4.3. Implications*

We seek to develop EHR-based dementia risk scores to support the future screening of dementia in clinical settings that include both BAs and WAs. That is, this work does not attempt to diagnose patients using EHRs but to generate risk scores that can be used to flag patients who may benefit from engaging in a clinical diagnostic process (e.g., completing brief cognitive screening or functional status assessments with health technicians followed by assessments by specialists). Other researchers have noted that PPV and NPV are better at evaluating a screening test in clinical practice than sensitivity and specificity [38]. Our model generated a very high NPV at the 90th percentile for both BA Veterans (0.98 [0.96, 0.99]) and WA Veterans (0.98 [0.94, 0.99]). These findings are similar to the NPVs reported with the eRADAR tool in an EHR sample that was 89% WAs [1] but are higher than the NPV reported by Yadgir et al. (i.e., 0.93) [35]. The PPV in our study was low for both BA Veterans (0.26 [0.21,0.31]) and WA Veterans (0.15 [0.12,0.20]) at the 90th percentile cutoff. Practically, this means that, at that threshold, about a quarter of the BAs and a seventh of the WAs who were flagged by our model as having potential dementia would actually have dementia according to our manual chart reviews. In contrast, Yadgir et al. achieved PPVs greater than 0.4, but to do so, they applied threshold cutoffs higher than 0.8; this meant that they obtained a high true positive rate at the expense of low sensitivity, which is not optimal as a screening instrument given the high cutoff scores [35]. Our algorithms compare very favorably to the eRADAR tool for dementia, which had a PPV of 0.115 in a research setting and 0.020 to 0.048 in patients [1,34]. Our PPV is similar

or superior to the rates of standard screening methods for cancers like mammograms or colonoscopy reviewed in [1]. However, cancer screening is often followed by more definitive tests, such as ultrasound and/or biopsies, and thus, low PPVs in screening tests may be acceptable. Likewise, we envision the implementation of our SVM model in clinical workflows as part of a multistage screening that would likely include diagnostic tests and clinical assessments.

### 4.4. Limitations and Future Work

The VA patient population skews heavily toward older males, and our training and test data thus had a low percentage of females; that may limit the generalizability of our final ML models outside VHA, though we expect that the same steps could be applied to generate risk scores within other health care systems with more females. We also acknowledge that we cannot clinically diagnose dementia based on manual chart reviews and that, in some cases, we were unable to retrospectively apply the newest AD criteria (NIA-Reagan) [39] due to insufficient information. This parallels the problem of underdiagnosis, as our reviewers were unable to assign a diagnosis if signs and symptoms relevant to impairment were not documented in clinical notes. This may have also led to a low level of dementia prevalence (i.e., per chart review), and a low prevalence of any condition leads to models with high NPVs and low PPVs. It is possible, therefore, that our model may catch signs of dementia that cannot be captured by a manual chart review, which means that our model may perform better when compared to more accurate diagnostic standards, like in-person expert diagnoses or neuropathological assessments; this represents a promising area for future research.

We recognize that future studies need to assess the portability of the ML models that we have developed. Not all EHRs have notes available to researchers (due to privacy issues), and in those instances, researchers are unable to leverage the full benefit of our models' ability to search both structured and unstructured data. Future studies should investigate how other ML methods, like deep learning approaches, might improve the detection of undiagnosed dementia and solicit input from BA stakeholders regarding model implementation in clinical processes. Finally, we expect that the implementation of our EHR-based risk scores will significantly increase the number of BA patients who are referred to specialists for dementia diagnosis, but future studies should investigate whether the implementation of our risk scores sufficiently improves the identification of dementia in clinics that serve both BA and WA Veterans.

**Supplementary Materials:** The following supporting information can be downloaded at https:// www.mdpi.com/article/10.3390/bdcc7040167/s1: Figure S1: Distribution of scores by dementia status and race (BA: Black American; WA: White American) for Veterans in the Training Sample (n = 5000 in each dementia status group for each race); Figure S2: Distribution of scores by race (BA: Black American; WA: White American) for Veterans in the Validation Sample (n = 10,000 for each race; n = 600 had chart review for each race); Figure S3: Receiver operating characteristic (ROC) curves based on observed values of sensitivity and specificity for Black American (BA) and White American (WA) Veterans who had chart reviews; Table S1a: Top 20 most important variables in the support vector machine (SVM) model for Black American (BA) Veterans, ranked by absolute value of variable weight in the model; Table S1b: Top 20 most important variables in the support vector machine (SVM) model for White American (WA) Veterans, ranked by absolute value of variable weight in the model.

**Author Contributions:** Conceptualization, D.W.T., Q.T.Z. and S.P.M.; formal analysis, K.B., K.T., K.W. and S.P.M.; funding acquisition, A.S.-D., D.W.T. and Q.T.Z.; investigation and validation, A.T., D.W.T. and K.C.; methodology, D.W.T., Q.T.Z. and Y.S.; project administration, D.W.T. and Q.T.Z.; software, Y.S.; visualization, K.B. and S.P.M.; writing—original draft preparation and literature review, A.S.-D. and S.P.M.; writing—review and editing, A.S.-D., D.W.T., K.B., K.T., K.W., Q.T.Z., S.P.M. and Y.S. All authors have read and agreed to the published version of the manuscript.

**Funding:** This work was supported by NIA R56 AG059739 and was supported in part by the U.S. Department of Veterans Affairs Office of Research and Development Biomedical Laboratory Research Program.

**Institutional Review Board Statement:** All procedures were approved by the IRBs at our respective institutions and were performed with an IRB-approved waiver of consent.

**Data Availability Statement:** The VA EHR data reside in VINCI behind VA firewalls. VA-approved investigators can access the data. SVM algorithms can be made available to interested qualified investigators.

**Conflicts of Interest:** The authors have no conflict of interest to report.

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
