# Peer review of "Identifying Probable Dementia in Undiagnosed Black and White Americans Using Machine Learning in Veterans Health Administration Electronic Health Records"

_2504-2289, doi:10.3390/bdcc7040167_

Round 1

Reviewer 1 Report

I read this paper with interest, and it was a good opportunity to test whether the deep learning model presented by the author can actually be implemented. We believe that if we make improvements based on the points below, we will be able to provide greater help to our readers.

Lack of Clarity in Objectives:

The abstract begins by discussing the potential of machine learning (ML) in reducing dementia underdiagnosis in electronic health records (EHRs) but fails to explicitly state its research objectives. It is crucial to clearly articulate the primary research questions and hypotheses to guide the study's design and interpretation.

Ambiguity in Model Development:

The abstract mentions the development and validation of ML models for assigning race-specific dementia risk scores using structured and unstructured VA EHRs. However, it lacks specificity regarding the choice of ML algorithms, feature selection processes, and validation techniques employed. A more detailed description of these aspects is essential for scientific reproducibility.

Lack of Transparency in Data:

The abstract mentions the use of training and validation samples for Black Americans (BAs) and White Americans (WAs), but it does not provide details about the characteristics of the data used, such as the demographics, data preprocessing, and any potential biases or limitations in the EHR data. Transparent reporting of data sources and preprocessing steps is crucial for assessing the study's validity.

Equivalence Mapping and Topic Modeling:

The abstract briefly mentions the use of equivalence mapping and topic modeling in model development, but it lacks clarity on how these techniques were implemented and their relevance to the research question. The abstract should provide a concise explanation of their roles and contributions to the study.

Limited Information on Model Performance:

While the abstract presents some performance metrics (e.g., AUC, NPV, PPV), it does not provide a comprehensive assessment of model performance, including sensitivity, specificity, and overall predictive accuracy. A more thorough evaluation of the models' performance is necessary to assess their clinical utility.

Implications and Generalization:

The abstract suggests that race-specific ML models can help identify undiagnosed dementia in minority patients. However, it does not discuss the potential implications of these findings for clinical practice or the generalizability of the models to different healthcare settings and populations.

additional issues:

While the abstract outlines a study with the potential to address a significant healthcare issue, it lacks clarity and specificity in describing its methodologies and results. To enhance scientific rigor and readability, the authors should provide a more detailed account of their research methods, data, and model performance metrics in the main text of the article. This will enable a more thorough assessment of the study's contributions to the field of dementia diagnosis and healthcare equity.

Moderate editing of English language required, 

Reviewer 2 Report

The manuscript, titled “Identifying Probable Dementia in Undiagnosed Black and White Americans Using Machine Learning in Veterans Health Administration Electronic Health Records,” presents machine learning models to predict race-specific dementia risk scores using. These authors developed a support vector machine model using structured and unstructured electronic health records from the Veterans Health Administration for 10,000 black and white Americans. Dementia is a major public health problem, affecting an estimated 50 million people worldwide and having a devastating impact on individuals, families, society and the economic burden on nations as a whole. Although the work has great potential for scholarly contribution, there are several concerns that should be addressed in order to be considered for publication.

** The following are my major concerns.

#1. The manuscript is difficult to understand and the text does not meet journal standards. Authors should exercise utmost care before submitting to the journal. See first line of summary. Authors must use correct capitalization in their texts. The text contains several grammatical errors. Some parts of the text are difficult to understand. In conclusion, their writing is premature.

#2. The authors should try other machine learning models such as Random Forests and XGBoost along with SVM as they claim that deep learning is the subject of future studies. Compare these models using performance metrics.

#3. Results and discussion need to be improved by providing a comparison with other methods (also their previous regression model) using standard ML performance metrics such as Accuracy, Precision, Matthews correlation etc., instead of providing the values reported in the published manuscripts.

The writing is premature and must be improved. 

Reviewer 3 Report

Reviewer understands that Shao et al. has presented a manuscript entitled "Identifying probable dementia in undiagnosed Black and White Americans using machine learning in Veterans Health Administration electronic health records". Reviewer has a few suggestions, and they request that the authors kindly answer all the questions by updating the requested details in their manuscript.

1) Throughout the manuscript, there are small mistakes with respect to superscript. i.e. in lines 36, 39 and 40, 90th, in line 123 to 124 similar mistakes are there. In lines 297, 299, and 304 similar mistakes are there.

2) Please mention your study's limiting or impacting factors/ parameters at previous stages and how you worked on them in your presented studies. What are the current limiting factors?

Round 2

Reviewer 2 Report

The authors have effectively addressed all of my concerns. I beleive that the manuscript is ready for publication in its current form.